# Cold Atmospheric Plasma-Treated PBS Eliminates Immunosuppressive Pancreatic Stellate Cells and Induces Immunogenic Cell Death of Pancreatic Cancer Cells

**DOI:** 10.3390/cancers11101597

**Published:** 2019-10-19

**Authors:** Jinthe Van Loenhout, Tal Flieswasser, Laurie Freire Boullosa, Jorrit De Waele, Jonas Van Audenaerde, Elly Marcq, Julie Jacobs, Abraham Lin, Eva Lion, Heleen Dewitte, Marc Peeters, Sylvia Dewilde, Filip Lardon, Annemie Bogaerts, Christophe Deben, Evelien Smits

**Affiliations:** 1Center for Oncological Research, University of Antwerp, 2610 Wilrijk, Belgium; 2Plasma, Laser Ablation and Surface Modelling Group, University of Antwerp, 2610 Wilrijk, Belgium; 3Laboratory of Experimental Hematology, University of Antwerp, 2610 Wilrijk, Belgium; 4Laboratory of General Biochemistry and Physical Pharmacy, Ghent University, 9000 Ghent, Belgium; 5Laboratory of Molecular and Cellular Therapy, Vrije Universiteit Brussel, 1090 Jette, Belgium; 6Department of Oncology, Multidisciplinary Oncological Center Antwerp, Antwerp University Hospital, 2650 Edegem, Belgium; 7Proteinchemistry, proteomics and epigenetic signaling group, University of Antwerp, 2610 Wilrijk, Belgium

**Keywords:** pancreatic cancer, pancreatic stellate cells, cold atmospheric plasma, immunogenic cell death, dendritic cells

## Abstract

Pancreatic ductal adenocarcinoma (PDAC) is one of the most aggressive cancers with a low response to treatment and a five-year survival rate below 5%. The ineffectiveness of treatment is partly because of an immunosuppressive tumor microenvironment, which comprises tumor-supportive pancreatic stellate cells (PSCs). Therefore, new therapeutic strategies are needed to tackle both the immunosuppressive PSC and pancreatic cancer cells (PCCs). Recently, physical cold atmospheric plasma consisting of reactive oxygen and nitrogen species has emerged as a novel treatment option for cancer. In this study, we investigated the cytotoxicity of plasma-treated phosphate-buffered saline (pPBS) using three PSC lines and four PCC lines and examined the immunogenicity of the induced cell death. We observed a decrease in the viability of PSC and PCC after pPBS treatment, with a higher efficacy in the latter. Two PCC lines expressed and released damage-associated molecular patterns characteristic of the induction of immunogenic cell death (ICD). In addition, pPBS-treated PCC were highly phagocytosed by dendritic cells (DCs), resulting in the maturation of DC. This indicates the high potential of pPBS to trigger ICD. In contrast, pPBS induced no ICD in PSC. In general, pPBS treatment of PCCs and PSCs created a more immunostimulatory secretion profile (higher TNF-α and IFN-γ, lower TGF-β) in coculture with DC. Altogether, these data show that plasma treatment via pPBS has the potential to induce ICD in PCCs and to reduce the immunosuppressive tumor microenvironment created by PSCs. Therefore, these data provide a strong experimental basis for further in vivo validation, which might potentially open the way for more successful combination strategies with immunotherapy for PDAC.

## 1. Introduction

Pancreatic ductal adenocarcinoma (PDAC) is a devastating disease with a five-year survival below 5%, making it one of the seven leading causes of cancer mortality in the world [1,2,3,4]. Given its rising incidence, it is estimated that, by 2030, PDAC will be among the top two most lethal cancers [5]. Only 10–20% of patients are eligible for curative surgical resection owing to the rapidly progressive nature of the tumor and, even with adjuvant chemotherapy, the median survival rate is only 20–23 months [1,2]. The only therapeutic options for the remaining 80–90% of patients are limited to chemo- and radiotherapy, which have minimal efficacy because of therapy resistance [3].

Although immunotherapy is considered to be a major breakthrough in cancer treatment, it has not yet achieved promising outcomes in PDAC. The ineffectiveness of immunotherapy may be explained by these tumors being non-immunogenic [6,7,8,9]. The immunosuppressive tumor microenvironment (TME) is believed to be a major underlying factor for immunotherapy failure. A hallmark of this TME is a desmoplastic reaction, which results in a dense fibrotic/desmoplastic structure surrounding the tumor. This dense stroma acts as a mechanical and functional shield, causing diminished delivery of systemically administered anticancer agents and immune cell infiltration, as a consequence of intratumoral pressure and low microvascular density, which results in therapy resistance [10,11,12,13]. The main orchestrators of this stromal shield are the activated pancreatic stellate cells (PSCs). These myofibroblast-like cells, also known as cancer-associated fibroblasts, enhance the development, progression, and invasion of PDAC through extensive crosstalk with pancreatic cancer cells (PCCs), resulting in reciprocal stimulation. Furthermore, PSC also directly influence immune cells by secreting immunosuppressive factors, like TGF-β [12,14]. Therefore, new treatment options that could overcome this stromal shield, and consequently increase tumor immunogenicity in PDAC, are necessary.

One way to enhance immunogenicity is by inducing immunogenic cell death (ICD), a form of cell death, which causes these dying cells to elicit an antitumor immune response [15]. Cancer cells undergoing ICD expose proteins on their surface and release immunogenic factors, so-called ‘damage-associated molecular patterns’ (DAMPs). Classically, there are three well-known DAMPs related to ICD. The first is surface-exposed calreticulin (ecto-CRT), which serves as an ‘eat me‘ signal. This marks tumor cells for engulfment by dendritic cells (DCs), which are professional antigen-presenting cells [16]. The second DAMP is adenosine triphosphate (ATP), secreted into the extracellular environment, serving as a chemoattractant for immune cells [17]. The third DAMP is high-mobility group box 1 (HMGB1), released into the extracellular milieu, which contributes to DC maturation [18,19]. Conversely, ICD is usually also accompanied by downregulation of the ‘don’t eat me’ signal CD47, which can inhibit phagocytosis of dying cancer cells [20]. Altogether, these signals stimulate DC, key players for initiating an adaptive immune response. Activated DC will lead to the development and activation of effector T cells, capable of specifically and systemically eradicating cancer cells, and of memory T cells, which provide long-term protection against cancer recurrence [21].

Several physical methods of cancer treatment, including radiotherapy, photodynamic therapy, and high hydrostatic pressure, are known inducers of ICD [22,23,24,25]. The induction of oxidative stress through the production of reactive oxygen species (ROS) is the common underlying factor of these therapies. In recent years, cold atmospheric plasma, which is a partially ionized gas consisting of a variety of reactive oxygen and nitrogen species (RONS), has emerged as a novel cancer treatment [26,27]. For simplicity, cold atmospheric plasma will be further referred to as ‘plasma’ in this paper. These RONS can be delivered directly to the tumor or indirectly through plasma-treated liquids [27]. Several studies have attributed the plasma-induced cancer cell death to the formation of exogenous and endogenous RONS, which lead to intracellular stress and ultimately cell death [27,28,29,30]. Therefore, we hypothesized that plasma could also be a potent inducer of ICD.

The aim of the present study is to evaluate the potency of plasma-treated phosphate-buffered saline (pPBS) as an anticancer modality to tackle PCCs and the immunosuppressive PSCs. Therefore, we evaluated the cytotoxic effect of pPBS treatment on both PCCs and the tumor-supportive PSCs. Additionally, we examined the immunogenicity of this cytotoxic effect on PCCs and PSCs based on the release of ICD markers and activation of DCs.

## 2. Results

### 2.1. pPBS Induces Cell Death in Both PCCs and PSCs

In order to initially determine a dose of pPBS treatment, which induces a significant amount of cell death in each cell line, we treated PCC and PSC lines with several dilutions of pPBS (25%, 37.5%, 50%, and 62.5%). After 48 h of treatment, we analyzed cell death with Annexin V (AnnV) and propidium iodide (PI) flow cytometric staining. All cell lines demonstrated a dose-dependent increase in AnnV−/PI+, AnnV+/PI+, and AnnV+/PI− cells, with a corresponding decrease in viable AnnV−/PI− cells (Figure 1a,b, Appendix A). MIA-Paca-2 cells were most sensitive to the treatment, followed by Capan-2. Therefore, these two cell lines were treated with the lowest concentration of pPBS for subsequent experiments compared with all other cell lines. Overall, PSC lines were significantly less sensitive to pPBS treatment compared with PCC lines (Figure 1c).

### 2.2. pPBS Induces ICD Markers on PCCs

Because therapy-induced tumor ICD is an important component to activate antitumor immunity, we investigated whether pPBS induces ICD in PCC and PSC lines. To this end, we measured the surface exposure of CRT as well as secretion of ATP and release of HMGB1 into the supernatant.

We observed a dose-dependent translocation of ecto-CRT in all PCC and two PSC lines after 48 h of pPBS treatment (Figure 2a, Appendix A). A strong translocation was detected for MIA-Paca-2 and Capan-2 cells with a mean of 20.1% and 10.5% ecto-CRT+ cells, respectively. Less pronounced, but still significant effects on the translocation were observed for PANC-1, BxPC3, hPSC128, and hPSC21 cells. Here, even the highest concentration of pPBS exposed not more than 7.5% ecto-CRT on the cell surface. No difference in ecto-CRT was observed for RLT-PSC cells.

Next, we measured extracellular ATP levels 4 h after pPBS treatment (Figure 2b). For two PCC lines, MIA-Paca-2 and PANC-1, accumulation of extracellular ATP up to five-fold from the untreated control was observed. Similar to ecto-CRT, the trend of secretion was dose-dependent. No significant accumulation was seen for the other cell lines.

On the basis of our previous cytotoxicity results, we chose one specific dose for every cell line to evaluate HMGB1 release. As indicated above, MIA-Paca-2 and Capan-2 were the most sensitive cell lines, and thus received a dose of 37.5% pPBS, as opposed to 50% pPBS for the other cell lines. pPBS treatment induced significant release of HMGB1 in all PCC lines, with a 1.32- to 1.79-fold increase compared with the untreated control. Interestingly, no significant release was detected in the PSC lines (Figure 2c). Additionally, we observed a significant downregulation of CD47 expression in all cell lines after pPBS treatment, except for Capan-2 and RLT-PSC (Figure 2d).

Collectively, our results show that plasma treatment via pPBS application is able to induce events that are characteristic of ICD in PCC. Importantly, pPBS-induced cell death in the PSC lines appears to be non-immunogenic owing to the absence of most DAMPs. For both MIA-Paca-2 and PANC-1, all four markers of ICD were significantly detected after pPBS treatment. The quantity of the examined markers was both dose and cell line dependent.

### 2.3. pPBS-Treated Cells are Phagocytosed by DCs

In view of the role of ecto-CRT as an ‘eat-me’ signal, we investigated the influence of pPBS-treated PCC and PSC on the phagocytotic capacity by immature DCs. Flow cytometric analysis revealed that pPBS-treated MIA-Paca-2, Capan-2, hPSC128, and hPSC21 were phagocytosed by immature DCs more efficiently than their untreated counterparts (Figure 3a,b, Appendix A). This phagocytotic capacity by DCs was significantly correlated (R = 0.786, p = 0.036) with the exposure of ecto-CRT on the cell surface of the cell lines after pPBS treatment (Figure 3c).

### 2.4. pPBS Treatment of PCC Increases Maturation of DCs without Affecting Their Viability

In order to initiate an effective adaptive immune response, the expression and release of DAMPs by dying tumor cells must be followed by DC phagocytosis and DC activation. The ability of DCs to initiate such an immune response depends on their maturation status upon activation. Therefore, we analyzed three different maturation markers on the cell surface of DC: CD80, CD83, and CD86. There was a clear donor-dependent upregulation of CD86 on viable DC after coculturing with pPBS-treated target cells (Figure 4a). This variability was detected both between the cell lines and between DC from different blood donors cultured with the same cell line. However, using DC from different donors in coculture with pPBS-treated MIA-Paca-2 and PANC-1 cells, there was a consistent and significant upregulation of CD86. The effect was less pronounced or undetectable for CD83 and CD80 maturation markers in all cell lines (Appendix A). Notably, pPBS treatment of DCs alone without target cells had no significant effect on the maturation status, meaning that the observed maturation effect was the result of tumor cells dying in an immunogenic way. Furthermore, we also checked the viability of the DCs in coculture. We could not detect any significant differences in DC viability after 48 h of coculture with pPBS-treated cells compared with coculture with untreated target cells. Addition of pPBS to monocultures of DCs also showed no significant differences in viability compared with their untreated counterparts (Figure 4b).

### 2.5. Secretion of Cytokines after pPBS Treatment

Mostly, maturation of DCs is associated with an increase in the production of proinflammatory cytokines. Therefore, we evaluated the cytokine production of TNF-α and IFN-γ by DCs in coculture with pPBS-treated PCCs and PSCs, which are both central players in the process of DC maturation and antitumoral immune responses. The interaction between DCs and pPBS-treated PCCs or PSCs induced the release of IFN-γ and TNF-α (Figure 5a,b). The release of both cytokines was significant for MIA-Paca-2 and PANC-1. In BxPC3 and Capan-2 cells, IFN-γ release was also significantly increased. In addition to these proinflammatory cytokines, we evaluated a well-characterized immunosuppressive cytokine, TGF-β, which is often released in the TME [31]. We observed a decrease in TGF-β release when DCs were cocultured with pPBS-treated BxPC3, hPSC128, and hPSC21 cells compared with cocultures with the untreated counterparts (Figure 5c).

## 3. Discussion

The purpose of this study was to investigate the ability of plasma treatment via pPBS to create a more immunogenic TME for PDAC by attacking both PSCs and PCCs and inducing ICD in PCCs. Figure 6 gives an overview of the immunogenic signals tested after pPBS treatment in four different PCC lines and three different PSC lines.

PDAC is known to have a low immunogenic TME profile and is often referred to as a ‘cold’ immunogenic tumor [32]. Because of its low immunogenicity, immunotherapy frequently fails in this type of tumor [6,7,33]. The dense stroma consisting of PSC surrounding the tumor is believed to be a major underlying factor involved in the failure of immunotherapy by acting as a physical barrier for drugs and immune cells [12,13]. Additionally, PSC secrete immunosuppressive factors, which prevent the development of effective immune responses [14]. Several studies showed that stromal depletion combined with immunomodulation resulted in better outcomes than immunomodulation alone in PDAC [34,35]. Therefore, we postulated that tumors can become immunogenically ‘hotter’ by destroying the tumor supporting PSC with pPBS treatment.

Although the cytotoxic effect of plasma has already been investigated in PCC lines [28,29,36,37] and a PSC line [37], we demonstrated the first use of pPBS to target the immunosuppressive PSC in PDAC and investigated its immunogenic potential. Treatment with pPBS induced non-immunogenic cell death in PSC, as seen by the lack of DAMP emission, except for ecto-CRT and CD47 expression, and no significant DC maturation. This is in line with the report of Gorchs et al. showing that cancer-associated fibroblasts in the lung do not undergo ICD after exposure to high dose radiotherapy [38]. Interestingly, secretion of the immunosuppressive TGF-β decreased in cocultures of DCs with pPBS-treated PSC lines, compared with their untreated counterparts. TGF-β plays a major role in immunosuppression within the TME and is often strongly secreted by PSC [39]. TGF-β is responsible for preventing immune cell infiltration into tumor tissue and promoting tumor cell proliferation [31,40,41]. Therefore, several ongoing efforts in this field are aimed at blocking TGF-β in the stroma in combination with anti-programmed death (PD)-1 immunotherapy for the treatment of different cancer types, including pancreatic cancer [31]. Similarly, we showed that pPBS treatment could kill PSCs and thereby disrupt the physical barrier, and additionally lower their immunosuppressive capacity. These findings show that plasma treatment can be beneficial in combination with immunotherapy for PDAC treatment.

PCCs were intrinsically more sensitive to pPBS treatment compared with PSCs. The delicate redox balance in PCCs may contribute to this observation. Cancer cells are characterized by increased production of ROS compared with normal cells, which promotes their tumorigenicity. This altered redox environment can increase their susceptibility to ROS-promoting therapies like pPBS by disturbing ROS homeostasis, resulting in lethal ROS levels and ultimately cancer cell death [42]. Furthermore, in contrast to PSCs, four signals, which play a key role in the immunogenic potential of ICD inducers, were identified after pPBS treatment in both MIA-Paca-2 and PANC-1 tumor cells. Both Lin et al. and Freund et al. showed a similar release of DAMPs by plasma treatment using different human and murine cancer cell lines [43,44,45,46]. Recently, Azzariti et al. also showed an increase of ecto-CRT and ATP in PANC-1 [47]. Our study is the first to evaluate phagocytosis of plasma-treated cancer cells by DCs and DC maturation, which are both needed to confirm the immunogenic profile of tumor cells [48,49]. Phagocytosis of cancer cells by DCs improved after pPBS treatment in all PCC lines and consistent upregulation of the maturation-associated marker CD86 on DCs was observed in cocultures with pPBS-treated MIA-Paca-2 and PANC-1 cells. Both cell lines highly express ecto-CRT and released ATP and HMGB1 after treatment, and showed a high downregulation in CD47 expression after treatment, resulting in more phagocytosis and DC maturation. Contrary to PSCs, we also observed an increased secretion profile of both TNF-α and IFN-γ in cocultures of DCs with pPBS-treated MIA-Paca-2 and PANC-1 cells. These data indicate a more immunogenic type of phagocytosis with higher production of proinflammatory cytokines, which is documented to lead to immunostimulatory clearance of tumor cells [20,50,51]. Furthermore, this complements our past study where mice, inoculated with a plasma-generated, whole-cell vaccine, were protected against live tumor challenge with melanoma cancer cells [52]. This strongly suggests that downstream of ICD, an adaptive immune response is triggered, which ultimately leads to the development of anti-tumor memory.

It has been shown that ATP could amplify the effects of other activators of DCs, such as TNF-α [53]. This could explain the lack of DC maturation after coculture with pPBS-treated BxPC3 and Capan-2 cells, as both cell lines did not release a significant amount of ATP after pPBS treatment, nor TNF-α in coculture with DCs. These observations further emphasize the importance of intrinsic differences between cell types and even cell lines when investigating the immunogenicity of treatment. Similar differences between tumor cell lines have been documented by Di Blasio et al [48]. Altogether, our data indicate that cocultures of pPBS-treated tumor cells and DCs are capable of releasing immunostimulatory signals in the TME, suggesting the induction of a more pronounced antitumoral immune response.

As DCs are important players in inducing specific antitumor immune responses and are potentially present in the TME, it is also important to identify the direct effects of plasma treatment on this subtype of immune cells [54,55]. We showed that pPBS treatment had no effect on the viability of DCs in monoculture or in coculture with PSCs and PCCs. A previous study shows that plasma induces apoptosis in PMBC in general [56]. However, when looking more specifically into the subpopulations of PBMC, Bekeschus et al. showed that monocytes are more resistant to plasma treatment. This could be because of a stronger antioxidant defense system in phagocytes, such as monocytes, macrophages, and DCs, which, under physiological conditions, protects them against self-production of ROS during oxidative burst [57].

In this study, we demonstrated that pPBS treatment may be an effective anticancer immunotherapeutic modality for PDAC by simultaneously attacking both PCCs and PSCs. Consequently, the physical barrier of PSCs might be disrupted, which could lead to more infiltration of immune cells. Together with the induction of ICD in PCC and the reduction of immunosuppressive cytokines released by PSCs, these results may potentially open the way for more successful combination strategies with immunotherapy in PDAC. In a next step, implementation of an in vivo model would be warranted. Nevertheless, we are convinced that our data have a high translational value, though extrapolation of in vitro cell line studies to the clinic should be considered with caution. Therefore, we believe that our experiments provide a strong experimental basis for further development of an in vivo model, which can make the translational value even stronger towards a clinical setting.

## 4. Materials and Methods

### 4.1. Cell Lines and Cell Culture

The human PCC lines MIA-Paca-2, PANC-1, BxPC3, and Capan-2 (ATCC) were used in this study. MIA-Paca-2 and PANC-1 cells were cultured in Dulbecco’s Modified Eagle Medium (DMEM; Life Technologies, 10938, Merelbeke, Belgium) supplemented with 10% fetal bovine serum (FBS, Life Technologies, 10270-106, Merelbeke, Belgium), 1% penicillin/streptomycin (Life Technologies, 15140), and 2 mM L-glutamine (Life Technologies, 25030). Capan-2 and BxPC3 cells were cultured in Roswell Park Memorial Institute (RPMI) 1640 medium (Life Technologies, 52400), supplemented as described above. The human PSC lines hPSC21, hPSC128 (established at Tohoku University, Graduate School of Medicine, kindly provided by Prof. Atsushi Masamune), and RLT-PSC (established at the Faculty of Medicine of the University of Mannheim, kindly provided by Prof. Ralf Jesenofsky) were used, all cultured in DMEM-F12 (Life Technologies, 31330), supplemented as described above [58,59]. Cells were maintained in exponential growth phase at 5% CO_2_ in a humidified incubator at 37 °C. Cell cultures were tested regularly for absence of mycoplasma contamination using the MycoAlert detection kit (Lonza, LT07, Verviers, Belgium)

### 4.2. Treatment of PCC and PSC with Cold Atmospheric Plasma

Cells (2 × 10^4^ cells per mL) were treated indirectly with cold atmospheric plasma generated using the atmospheric pressure plasma jet kINPenIND^®^ (Neoplas Tools). Argon gas is used in this setting as feeding gas [60]. Then, 2 mL of PBS was treated with one standard liter per minute (slm) gas flow rate at a gap distance of 6 mm for 5 min. This 100% plasma-treated PBS (pPBS) was further diluted in PBS to final concentrations of 12.5%, 25%, 37.5%, 50%, 62.5% pPBS, which was then directly added in a 1/6 dilution in the media to the cells. Untreated PBS is used as a vehicle control for all experiments.

### 4.3. Analysis of Cytotoxicity and ICD Markers

Forty-eight hours after treatment, cells were harvested and incubated with 5% normal goat serum (NGS, Sigma-Aldrich, G9023, Overijse, Belgium), followed by washing and incubation with an Alexa Fluor 488-conjugated anti-CRT (Abcam, ab196158) antibody for 40 min. Prior to analyzing the samples, the cells were stained with Annexin V (BD, 550474) and PI (BD, 556463) to distinguish between early apoptotic and necrotic cells. The percentage of cytotoxicity presents [%AnnV+PI- + %AnnV-PI+ + %AnnV+PI+]. The surface expression of CRT was analyzed on non-permeabilized cells (PI-). For every sample, an isotype control was used (Abcam, 199091). Flow cytometric acquisition was performed on an AccuriTM C6 instrument (BD). Extracellular ATP was measured in conditioned media (supplemented with heat inactivated FBS) 4 h after treatment via ENLITEN^®^ ATP assay system, according to the manufacturer’s protocol (Promega, FF2000). The bioluminescent signal was measured using a VICTOR^TM^ plate reader (PerkinElmer). Release of HMGB1 was analyzed 48 h after treatment in the conditioned media using an enzyme-linked immunosorbent assay (IBL, ST51011). The absorption was measured using an iMARKTM plate reader (Bio-rad). Surface expression of CD47 (BD, 556046) was analyzed on non-permeabilized cells (7-AAD^-^, Biolegend, 420404), 48 h after treatment. Flow cytometric acquisition was performed on a CytoFLEX (Beckman Coulter) instrument.

### 4.4. In Vitro Generation of Human Monocyte-Derived DCs

Human peripheral blood mononuclear cells (PBMC) were isolated by LymphoPrep gradient separation (Sanbio, 1114547) from a buffy coat of healthy donors (Ethics Committee of the University of Antwerp, reference number 14/47/480) isolated from adult volunteer whole blood donations (supplied by the Red Cross Flanders Blood service, Belgium). Monocytes were isolated from PBMC using CD14 microbeads according to the manufacturer’s protocol (Miltenyi, Biotec, 272-01). Purity after isolation was >90%. After isolation, CD14+ cells were plated at a density of 1.25–1.35 × 10^6^ cells per mL in RPMI-1640 supplemented with 2.5% human AB (hAB, Sanbio, A25761) serum, 800 U/mL granulocyte-macrophage colony stimulating factor (GM-CSF; Gentaur, 04-RHUGM-CSF), and 20 ng/mL interleukin (IL)-4 (Miltenyi, Biotec, 130-094-117) at day 0, as described before [61]. Immature DCs were harvested on day 5.

### 4.5. Coculture of DCs and Tumor Cells

In order to measure the maturation and phagocytotic capacity of the immature DCs, a flow cytometric assay was used. To make a distinction between target and effector cells, they were both stained with a different fluorescent dye prior to coculturing. Labeling of immature DCs was performed as described before with minor adjustments [62]. Briefly, immature DCs were labeled with 2 µM of violet-fluorescent CellTracker Violet BMQC dye (Invitrogen, C10094, Bleiswijk, Netherlands) at a concentration of 1 × 10^6^ cells per mL at 37 °C. PCCs and PSCs were labeled with the green fluorescent membrane dye PKH67 (Sigma Aldrich, MIDI67). Labeling of tumor cells with PKH67 was carried out according to the manufacturer’s instructions and performed before pPBS treatment. Four hours after pPBS treatment, effector and target cells were cocultured at a 1:1 effector/target (E/T) ratio. Forty-eight hours later, supernatant was collected and stored at –20 °C for future analysis. Cells were collected and used immediately for flowcytometric detection of DC maturation markers and phagocytosis. Expression of CD80 (Biolegend, 400150, San Diego, CA, USA), CD86 (BD, 557872), and CD83 (BD, 551073) maturation markers was measured on the violet+ DC population. For every specific maturation marker, an isotype control was used (BD, 555751; BD, 557872; Biolegend, 305232). Difference in mean fluorescence intensity (ΔMFI) was calculated to evaluate target upregulation after treatment. ΔMFI represents [(MFI staining treated–MFI isotype treated)–(MFI staining untreated–MFI isotype untreated)]. Phagocytosis of PKH67+ tumor cells by violet-labeled DCs was expressed as %PKH67+violet+ cells within the violet+ DC population. Acquisition was performed on a FACSAria II (BD). Data analysis was performed using FlowJo v10.1 software (TreeStar).

### 4.6. Cytokine Secretion Profile

Secreted cytokines in cocultures of pPBS-treated target cells and immature DCs were analyzed using electrochemiluminescence detection on a SECTOR3000 (MesoScale Discovery/MSD) using Discovery Workbench 4.0 software, as previously described [63]. The human cytokine panel included IFN-γ, TNF-α, and TGF-β. Standards and samples were measured in duplicate and the assay was performed according to the manufacturer’s instructions.

### 4.7. Statistical Analysis

Prism 8.02 software (GraphPad) was used for data comparison and graphical data representations. SPSS Statistics 25 software (IBM) was used for statistical computations. The non-parametric Kruskal–Wallis test was used to compare means between more than two groups. The nonparametric Mann–Whitney U test was used to compare means between two groups. Spearman’s rank correlation coefficient was used to calculate the correlation between two variables. *p*-values < 0.05 were considered statistically significant.

## 5. Conclusions

We conclude that plasma treatment via pPBS can attack both the PCCs and the PSCs. These data show that pPBS has the potential to induce ICD in PCCs and to reduce the immunosuppressive tumor microenvironment created by PSCs. Altogether, these results might potentially open the way for more successful combination strategies with immunotherapy for the treatment of PDAC.

## Figures and Tables

**Figure 1 cancers-11-01597-f001:**
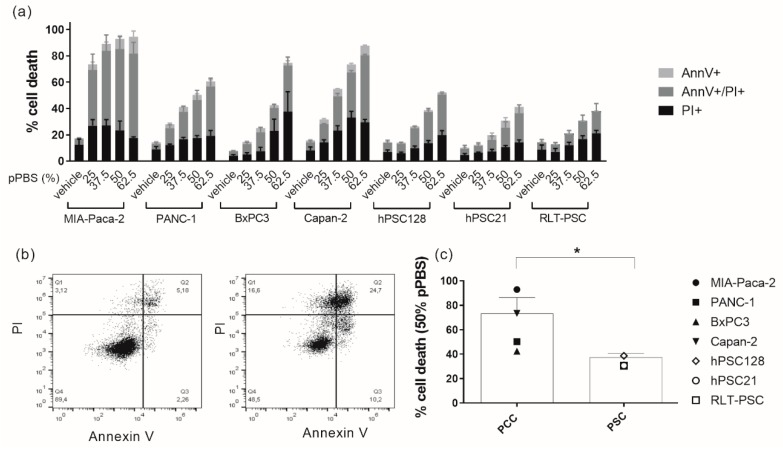
Sensitivity of pancreatic cancer cell (PCC) lines and pancreatic stellate cell (PSC) lines to different doses of plasma-treated phosphate-buffered saline (pPBS) treatment. (**a**) Percentage of cytotoxicity 48 h post pPBS treatment in four different PCC lines (MIA-Paca-2, PANC-1, BxPC3, Capan-2) and three different PSC lines (hPSC128, hPSC21, RLT-PSC). Subdivisions in the percentage Annexin V+, PI+, and double positive cytotoxic cells are made. (**b**) Dot plots showing the flow cytometric analysis of Annexin V and PI staining after 25% pPBS treatment in MIA-Paca-2 (right) compared with the untreated control (left): Q1 = AnnV−/PI+; Q2 = AnnV+/PI+; Q3 = AnnV−/PI−; Q4 = AnnV+/PI−. Representative dot plots for all other cell lines are presented in Appendix A. (**c**) The difference in sensitivity after 48 h of 50% pPBS treatment for means of all PCC lines and all PSC lines. Graphs represent mean ± SEM of ≥3 independent experiments. * *p* < 0.05.

**Figure 2 cancers-11-01597-f002:**
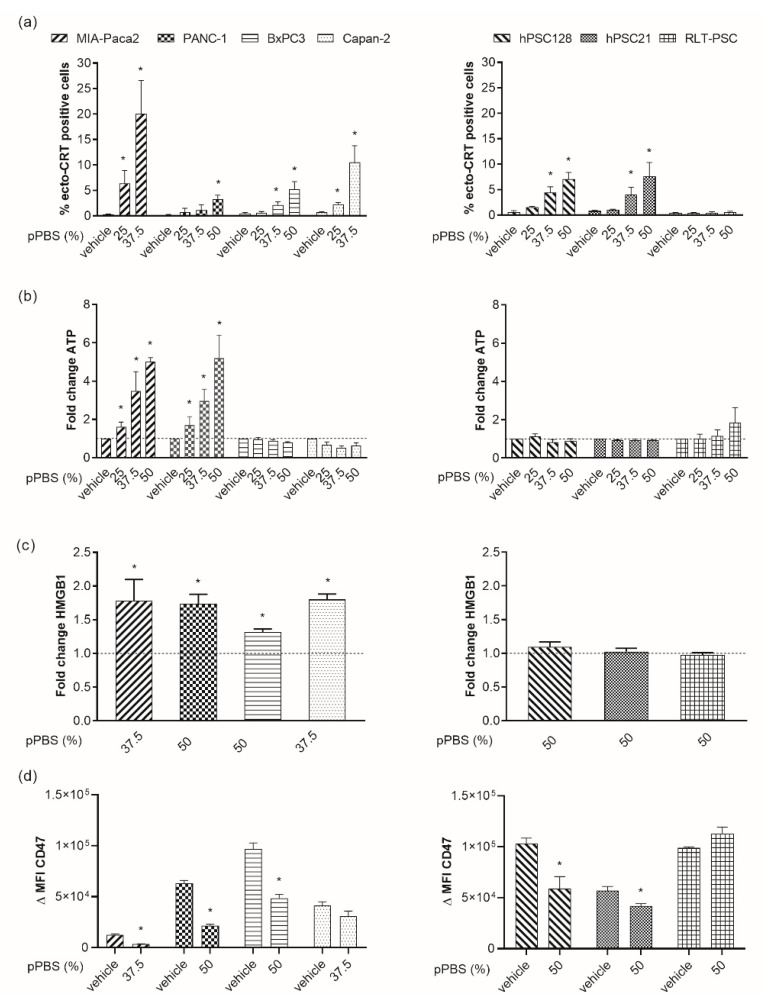
Release of immunogenic cell death (ICD) markers after pPBS treatment. (**a**) Percentage of surface-exposed calreticulin (ecto-CRT) positive cells after increasing the dose of pPBS treatment (25%, 37.5%, 50% pPBS). (**b**) Adenosine triphosphate (ATP) secretion 4 h post treatment in the supernatant. (**c**) High-mobility group box 1 (HMGB1) secretion 48 h post pPBS treatment in supernatant. These data demonstrate the fold change of ATP secretion (ng/mL range) against the untreated control. (**d**) Difference in mean fluorescence intensity (ΔMFI) of CD47 after 48 h of pPBS treatment. ΔMFI represents [(MFI staining treated–MFI isotype treated)–(MFI staining untreated–MFI isotype untreated)]. Different concentrations of pPBS treatment are used (25%, 37.5%, 50% pPBS). In the left graphs, four different PCC lines are represented (MIA-Paca-2, PANC-1, BxPC3, Capan-2), and in the right graphs, three different PSC lines are represented (hPSC128, hPSC21, RLT-PSC). Graphs represent mean ± SEM of ≥ 3 independent experiments. * *p* < 0.05 significant difference compared with untreated conditions.

**Figure 3 cancers-11-01597-f003:**
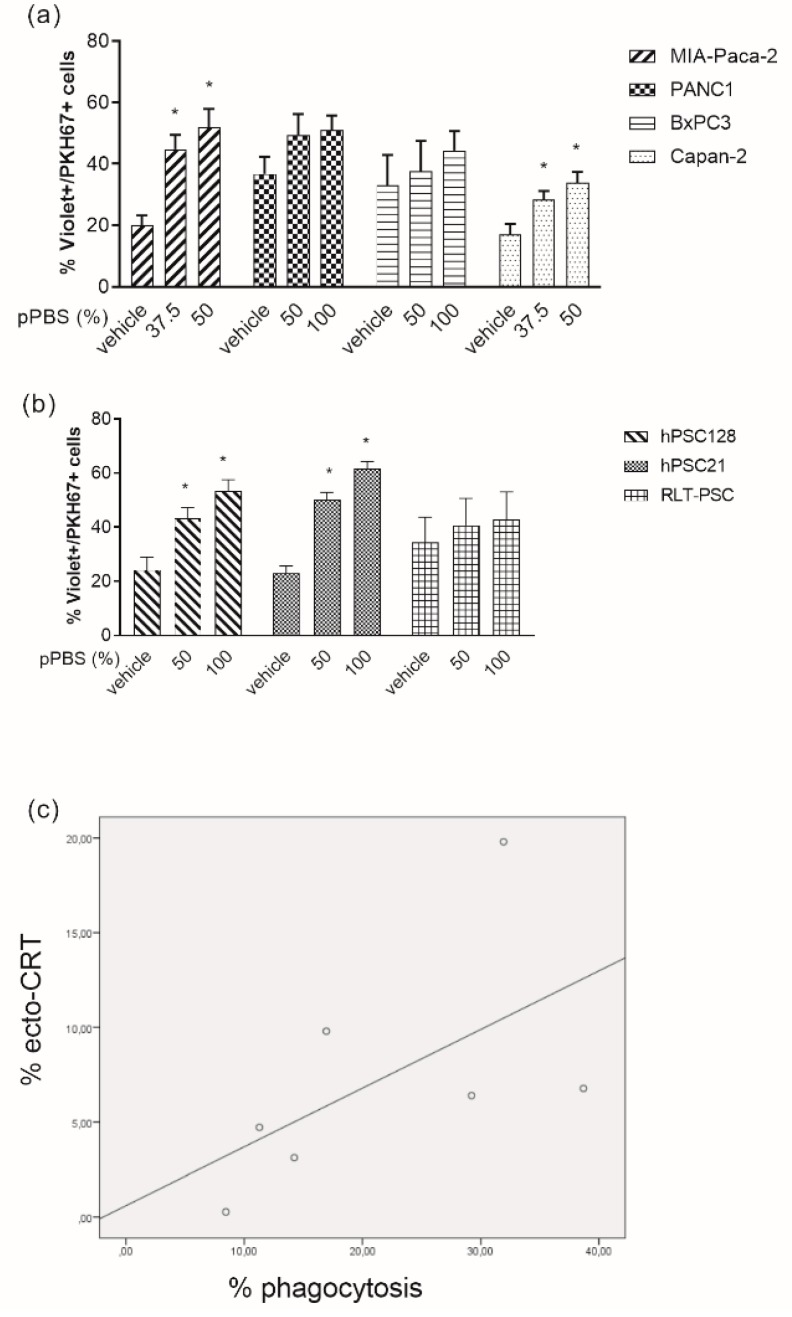
Phagocytosis of pPBS-treated PCCs and PSCs by immature dendritic cells (DCs). (**a**) Percentage of phagocytosis of four different PCC lines (MIA-Paca-2, PANC-1, BxPC3, Capan-2) and **(b**) three different PSC lines (hPSC128, hPSC21, RLT-PSC), with increasing dosage of pPBS treatment. Phagocytosis of PKH67+ tumor cells by violet-labeled DC is expressed as the %PKH67+violet+ cells within the violet+ DC population. (**c**) Correlation between exposure of ecto-CRT and phagocytotic capacity of DCs in the seven cell lines (R = 0.786, p = 0.036). Graphs represent mean ± SEM of ≥3 independent experiments. * *p* < 0.05 significant differences compared with untreated control.

**Figure 4 cancers-11-01597-f004:**
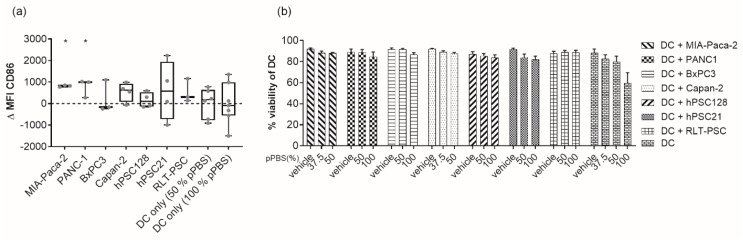
Maturation and viability of DCs after coculture with pPBS-treated PSCs and PCCs. (**a**) Box plot from minimum to maximum value of ΔMFI of the maturation marker CD86. CD86 expression is examined on immature DCs after 48 h of coculture of pPBS-treated PCCs and PSCs (effector/target (E/T) ratio, 1:1), and after pPBS treatment on immature DCs without coculture using flow cytometry. ΔMFI represents [(MFI staining treated–MFI isotype treated)–(MFI staining untreated–MFI isotype untreated)]. Treatment of 50% pPBS is used for MIA-Paca-2 and Capan-2, while treatment of 100% pPBS is used for PANC-1, BxPC3, hPSC128, hPSC21, and RLT-PSC. Every dot represents a different healthy donor and ≥3 donors were used per cell line. * *p* < 0.05 significant differences compared with untreated control. (**b**) Percentage of viability of DCs after 48 h coculture with pPBS-treated PCC lines (MIA-Paca-2, PANC-1, BxPC3, Capan-2) and PSC lines (hPSC128, hPSC21, RLT-PSC) or pPBS treatment alone. Graph represent mean of ± SEM of ≥3 independent experiments with different donors. * *p* < 0.05 significant differences compared with untreated control.

**Figure 5 cancers-11-01597-f005:**
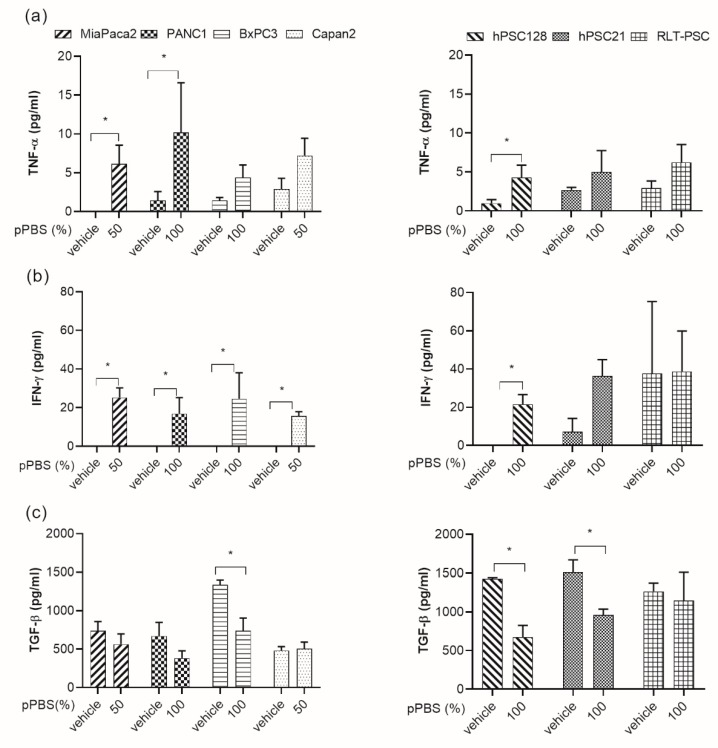
Cytokine profile released by DCs in coculture with pPBS-treated PCCs and PSCs. Graphs show the concentration of TNF-α (**a**), IFN-γ (**b**), and TGF-β (**c**) released in coculture of DCs with pPBS-treated PCCs (left) and PSCs (right) after 48 h. Graphs represent mean of ± SEM of ≥3 independent experiments with different donors. * *p* < 0.05 significant differences compared with untreated control.

**Figure 6 cancers-11-01597-f006:**
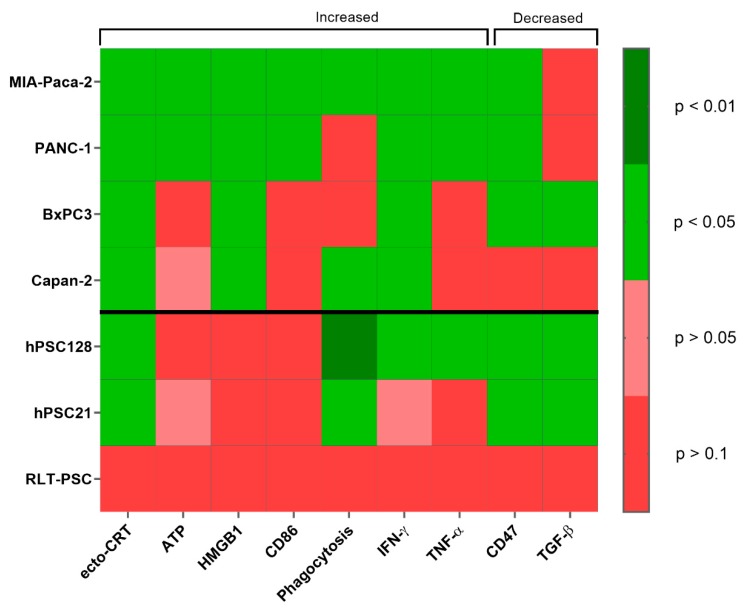
Overview of the *p*-values for all immunogenic signals tested. *p*-values are represented in a heatmap for the signals tested in previous experiments for both PCCs and PSCs. *p*-values are calculated using the Kruskall–Wallis or Mann–Whitney U test and are significant when <0.05. Treated conditions for ecto-CRT, ATP, HMGB1, CD86, phagocytosis, IFN-γ, and TNF-α are significantly increased compared with untreated controls. Treated conditions for CD47 and TGF-β are significantly decreased compared with untreated control.

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
