# Peer review of "Cold Atmospheric Plasma-Treated PBS Eliminates Immunosuppressive Pancreatic Stellate Cells and Induces Immunogenic Cell Death of Pancreatic Cancer Cells"

_cancers, 2019, doi:10.3390/cancers11101597_

Round 1

Reviewer 1 Report

WHile the reported in vitro data are of potentially high translational value they require confirmation in an in vivo system such as mouse xenografts, genetically engineered cancer codel or chemically induced cancer model.

Reviewer 2 Report

The manuscript "Cold atmospheric plasma eliminates immunosuppressive pancreatic stellate cells and induces immunogenic cell death of pancreatic cancer cells" addresses an important and interesting topic highlighting the emerging relevance to attack cancer cells and simultaneously to modulate TME. 

Major points:

1. About phagocytosis assay results: it is well known that the anti-phagocytic molecule CD47 significantly contributes to damaged cell engulfment and elimination by DCs and generally the balance CD47/ecto-CRT is the real determinant of this process. Moreover, the down-regulation of CD47 during ICD is documented by several studies:

1.Montico B, Lapenta C, Ravo M, Martorelli D, Muraro E, Zeng B, Comaro E, Spada M, Donati S, Santini SM, Tarallo R, Giurato G, Rizzo F, Weisz A, Belardelli F, Dolcetti R, Dal Col J. Exploiting a new strategy to induce immunogenic cell death to improve dendritic cell-based vaccines for lymphoma immunotherapy. Oncoimmunology. 2017 Jul 31;6(11):e1356964. doi: 10.1080/2162402X.2017.1356964.

2. Garg AD, Romano E, Rufo N, Agostinis P. Immunogenic versus tolerogenic phagocytosis during anticancer therapy: mechanisms and clinical translation. Cell Death Differ. 2016 Jun;23(6):938-51. doi: 10.1038/cdd.2016.5. Epub 2016 Feb 19. Review. PubMed PMID: 26891691; PubMed Central PMCID: PMC4987738.

3. Adkins I, Sadilkova L, Hradilova N, Tomala J, Kovar M, Spisek R. Severe, but not mild heat-shock treatment induces immunogenic cell death in cancer cells. Oncoimmunology. 2017 Mar 31;6(5):e1311433. doi: 10.1080/2162402X.2017.1311433. eCollection 2017. PubMed PMID: 28638734; PubMed Central PMCID: PMC5467989.

.....

It cuold be very interesting to evaluate CD47 in the cell lines after pPBS treatment and to understand if where the phagocytic activity of DCs did not significantly increase it is due to the balance CD47/ecto-CRT.

2. For a complete evaluation of DC maturation it can be helpful the analysis of other cytokines involved in ICD such as IL-6 and IL-1ß and the evaluation of MHC-II expression

Reviewer 3 Report

Loenhout and co-authors have presented a very detailed study on the effects of pPBS by checking the inhibition of PSC and PCC simultaneously. They also claimed pPBS treated PCC expressed and release DAMP which leads to ICD and also highly phagocytized by DC. This work will aid the understanding of the effects of pPBS made from cold plasma on PSC and PCC and how these may be used to benefit for PDAC treatment. I think this is a very thorough piece of work but I believe it would have benefited from the inclusion of fresh PBS (untreated) as main control and I would recommend this for further studies going forward. I believe the manuscript warrants publication with some major revisions. Comments and queries can be found below. 

The title of the manuscript is not appropriate according to the content. The whole experiments were performed using pPBS, while the title mentioned: “Cold atmospheric plasma eliminates…..” The title needs to be rewritten with the inclusion of pPBS in it. Line 301 --> “Cold atmospheric plasma generated using … feeding gas”. The line is incomplete and needs to be rewritten. Line 305 --> “…added in a 1/6 dilution in the media to the cells.” Does the author use only PBS as a control for the experiments? By doing 1/6 dilution of pPBS in media, the percentage of FBS in the medium will also change which leads to alteration in growth and other properties of the cells also. Authors strongly recommend adding pPBS only group also in all experiments. Line 327 --> 1640 RPMI must be replaced by RPMI-1640. Figure 1 --> Why author specifically showed only MIA-Paca-2 contour graph after 25% pPBS treatment? It is recommended to add best dose (i.e., 62.5% pPBS) in all 5 cell lines by using dot plots of FACS. Figure 1 --> Y-axis mentioned “% cytotoxicity”. The apoptosis/necrosis are essential to keep cells in homeostasis. One cannot say % cytotoxicity for that even in general term. It is suggested to make them separately and mention “Apoptosis or Necrosis” separately in Y-axis. Page 8 --> The figure is without legend. Mention it at an appropriate place.

Round 2

Reviewer 1 Report

Since the authors do not appear to have the resources and/or expertise to strengthen their in vitro data by in vivo experiments, the discussion and abstact should state thatfuture  invivo studies will need to verify their conclusions.

Author Response

Thank you for the effort of reviewing the manuscript.

Reviewer 2 Report

In my opinion the authors improved the manuscript and answered all the point requested.

I consider the paper acceptable for publication. 

Author Response

(The authors gave the same response as above.)

Reviewer 3 Report

The manuscript can be accepted now in its revised version.

Author Response

(The authors gave the same response as above.)
